# Effects of Chronic Exposure to Low Doses of Rotenone on Dopaminergic and Cholinergic Neurons in the CNS of *Hemigrapsus sanguineus*

**DOI:** 10.3390/ijms25137159

**Published:** 2024-06-28

**Authors:** Elena Kotsyuba, Vyacheslav Dyachuk

**Affiliations:** A.V. Zhirmunsky National Scientific Center of Marine Biology, Far Eastern Branch, Russian Academy of Sciences, 690041 Vladivostok, Russia; epkotsuba@mail.ru

**Keywords:** rotenone, neurotransmitters, dopamine, acetylcholine, crustacea

## Abstract

Rotenone, as a common pesticide and insecticide frequently found in environmental samples, may be present in aquatic habitats worldwide. Exposure to low concentrations of this compound may cause alterations in the nervous system, thus contributing to Parkinsonian motor symptoms in both vertebrates and invertebrates. However, the effects of chronic exposure to low doses of rotenone on the activity of neurotransmitters that govern motor functions and on the specific molecular mechanisms leading to movement morbidity remain largely unknown for many aquatic invertebrates. In this study, we analyzed the effects that rotenone poisoning exerts on the activity of dopamine (DA) and acetylcholine (ACh) synthesis enzymes in the central nervous system (CNS) of Asian shore crab, *Hemigrapsus sanguineus* (de Haan, 1835), and elucidated the association of its locomotor behavior with Parkinson’s-like symptoms. An immunocytochemistry analysis showed a reduction in tyrosine hydroxylase (TH) in the median brain and the ventral nerve cord (VNC), which correlated with the subsequent decrease in the locomotor activity of shore crabs. We also observed a variation in cholinergic neurons’ activity, mostly in the ventral regions of the VNC. Moreover, the rotenone-treated crabs showed signs of damage to ChAT-lir neurons in the VNC. These data suggest that chronic treatment with low doses of rotenone decreases the DA level in the VNC and the ACh level in the brain and leads to progressive and irreversible reductions in the crab’s locomotor activity, life span, and changes in behavior.

## 1. Introduction

Rotenone, being widely used as a pesticide and insecticide, may be found in aquatic habitats worldwide, including rivers, estuaries, and marine waters, where it has a potential impact on non-target aquatic species and associated communities [1]. Due to the currently high rates of use of rotenone, understanding its neural and behavioral effects is crucial. Studies have demonstrated that long-term exposure to low concentrations of this compound may be a kind of environmental factor that can induce changes characteristic of Parkinson’s disease (PD) in both vertebrates and invertebrates [2,3,4,5].

In mammals, PD is a disorder where a cascade of cellular changes leads to selective degeneration of dopaminergic neurons in the substantia nigra of the midbrain, aggregation of α-synuclein, mitochondrial abnormalities in neurons, neuroinflammation, and oxidative stress, which results in the progressive disorder of motor functions [6,7,8]. Furthermore, rotenone may also cause an imbalance between the dopaminergic and cholinergic systems, which also contributes to motor deficits [9,10,11,12,13,14].

Like that in mammals, the major manifestation of rotenone-induced Parkinsonism in invertebrates is the movement morbidity and degeneration of dopaminergic neurons [2,15,16,17]. In various invertebrate groups, chronic exposure to rotenone induces a reduction in locomotion and selective loss of the enzyme tyrosine hydroxylase (TH) responsible for dopamine (DA) production in DA-ergic neurons [3,18,19,20]. Dopamine has long been known as a key neuromodulator of motor circuits that is critical for the activation, maturation, and modulation of locomotor patterns across invertebrates and vertebrates [21,22,23].

However, despite the motor dysfunction occurring with a decrease in dopamine [5,24], little is known about the changes in *acetylcholine* (ACh) and in dopamine/ACh balance in invertebrates as a result of rotenone exposure. An analysis of the toxic effects of rotenone has shown that all invertebrates are sensitive to rotenone, but the range of doses that cause these effects varies widely depending on the concentration, duration of exposure, and the species being exposed [25]. Crustaceans generally exhibit a relatively higher tolerance to rotenone [26]. Due to the relative simplicity of their neural circuits and their easily identifiable behavior, these animals are used as model organisms for behavioral, physiological, biochemical, and molecular research and are considered as biomarkers for assessing effects caused by toxic agents [20,27,28]. The Asian shore crab, *Hemigrapsus sanguineus* (de Haan, 1835), an inhabitant of coastal and estuarine waters, is highly tolerant to a wide range of temperatures and salinities, which characterizes this species as highly adapted to changing environmental conditions. This crab may become a suitable model for the study of the neural organization of motor control and also cellular and molecular mechanisms of pathological processes in nervous tics, including those in PD. Although crabs are phylogenetically/evolutionarily distant from humans, the fundamental cellular processes, as well as some regulatory functions of neurotransmitters and signaling pathways, are, nevertheless, conserved between both organisms.

Furthermore, the fundamental aspects of locomotor behavior regulation in decapod crustaceans, as in other *arthropods*, are quite similar to those in vertebrates, in particular as regards the structural and functional organization of the ventral nerve cord (VNC) that controls most of the automated, rhythmically occurring processes [29,30].

Studies on crustaceans can help to identify the effects of rotenone on the activity of neurotransmitters that govern the motor function. These are also expected to clarify the factors underlying the specific molecular mechanisms leading to movement morbidity.

For this reason, the aim of the present study was to investigate the effects of rotenone poisoning on dopaminergic and cholinergic neurons and on the dynamics of their immunoreactivity in the CNS of Asian shore crab, *H. sanguineus*, in the case of their transition to motor dysfunction.

## 2. Results

### 2.1. Effects of Rotenone Exposure on Motor Behavior and Survival

#### 2.1.1. Open Field Test

To examine the effect of rotenone exposure on the locomotor activity of crabs *H. sanguineus*, the total distance that the crabs moved within 5 min was measured in an open field test. Intact crabs covered a distance of, on average, 429.071 ± 32.82 cm during the 5 min test (*n* = 6) (Figure 1). The locomotor activity of the rotenone-exposed groups significantly decreased in the open field test after 7, 14, and 21 days of rotenone treatment (291.333 ± 23.30; 250.667 ± 30.61, and 166.917 ± 19.90, respectively) vs. control animals (426.571 ± 22.6) (Figure 1).

#### 2.1.2. Survival Rate

The survival rates for the control and intact groups were 100% during a 3-week experimental period; however, crabs in the rotenone-treated group began to die from day 21 to day 28. All animals in this group died by day 28 (Figure 2).

### 2.2. Effect of Chronic Exposure to Low Doses of Rotenone on Dopaminergic and Cholinergic Immunoreactivity in the CNS of Hemigrapsus sanguineus

In the crabs *H. sanguineus*, neurons showing tyrosine hydroxylase-like immunoreactivity (TH-lir) and choline acetyltransferase-like immunoreactivity (ChAT-lir) were present in the CNS, but the intensity of their staining in the control crabs did not differ from that in the intact animals and remained unchanged throughout the experiment. However, TH-lir and ChAT-lir in certain regions differed between the control animals and those exposed to low doses of rotenone (Figure 3, Figure 4, Figure 5, Figure 6 and Figure 7).

#### 2.2.1. TH-lir after Chronic Exposure to Low Doses of Rotenone

In the control and rotenone-treated crabs, rare TH-lir neurons and fibers were identified throughout the brain and the ventral nerve cord (VNC) (Figure 3, Figure 4 and Figure 5), with the pattern of their labelling resembling that of the DA distribution previously described for the brain [31,32,33,34] and VNC of other crustaceans [30,32,33,34,35,36]. When describing the crab brain, which comprises a protocerebrum consisting of optic ganglia, a lateral protocerebrum, and a median protocerebrum [37], in this study, we focused only on the median brain (median protocerebrum, deutocerebrum, and tritocerebrum) and did not consider the lateral protocerebrum and optic ganglia in the eyestalk.

In the median protocerebrum of the control crabs, TH-lir was detected in medium-sized (21–40 μm) and small (12–18 μm) neurons of the anterior medial cell cluster 6 (Figure 3A–F).

**Figure 3 ijms-25-07159-f003:**
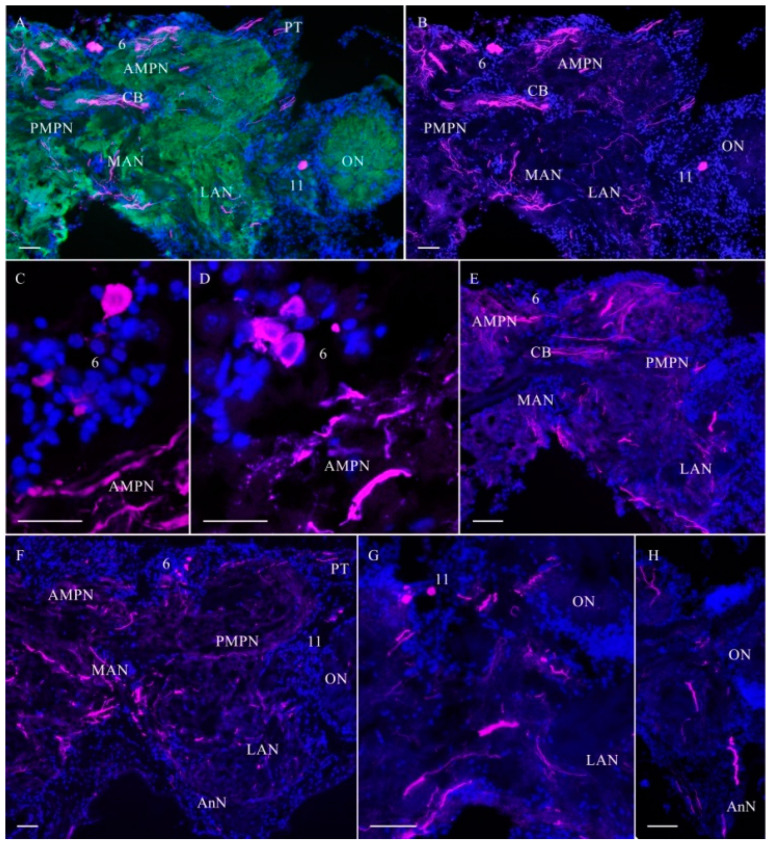
Distribution of tyrosine hydroxylase (TH)-lir in median brain of *H. sanguineus* in control. Horizontal sections through mid-dorsal planes of brain (**A**,**B**). Immunolabeling of median brain neuropils revealed by synapsin antibody (green) (**A**); TH-lir in neuronal clusters 6 and 11, AMPN, PMPN, CB, LAN, MAN, and PT, connection between protocerebrum and eyestalk (**B**). Neuronal cluster 6 contains medium-sized neurons with high TH-lir and small-sized neurons with low TH-lir (**C**,**D**). Horizontal sections through mid-ventral planes of brain showing TH-lir fibers in anterior and posterior medial protocerebrum neuropils, central body, and medial antenna neuropil (**E**). This view of almost maximum intensity projection shows labeled fibers running through MAN and LAN (**F**). Small TH-lir neurons of cluster 11 (**G**). TH-lir fiber running through AnN (**H**). The letter designations are as follows: AMPN, anterior medial protocerebral neuropil; PMPN, posterior medial protocerebral neuropil; ON, olfactory neuropil; CB, central body; LAN, lateral antenna l neuropils; MAN, medial antenna l neuropils; PT, protocerebral tract; AnN, antenna II neuropil; 6 and 11, cell clusters. Green color indicates synapsin; magenta, TH; blue, DAPI. Scale bars: 100 μm.

Pronounced TH-lir was also observed in the anterior (AMPN) and posterior (PMPN) medial protocerebral neuropils, in the central body (CB), and also in fibers of the protocerebral tracts (PT) (Figure 3A,B,E,F and Figure 4A). In the deutocerebrum, TH-lir was detected in few small (10–18 μm) and medium-sized (21–45 μm) cells in cluster 11 (Figure 3A,B,F,G). TH-lir fibers were also found in the medial antenna I neuropil (MAN) and lateral antenna I neuropils (LAN) (Figure 3A,B,E–H and Figure 4A). In the tritocerebrum, single TH-lir varicose fibers, which extended all over the length of antenna II neuropil to the VNC, were labeled intensely (Figure 3H).

After 7 days of rotenone exposure, a decline in TH-lir was observed in cell cluster 6 and a reduction in the intensity of labeling of neuritic processes, especially in MAN and PMPN (Figure 4B,C), vs. control.

**Figure 4 ijms-25-07159-f004:**
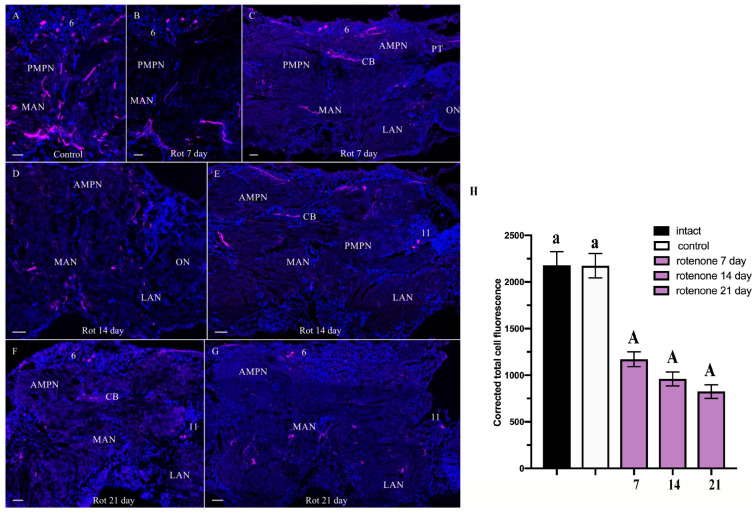
Changes in tyrosine hydroxylase (TH)-lir in median brain of control crabs *Hemigrapsus sanguineus* (**A**) and in rotenone-treated animals on days 7 (**B**,**C**), 14 (**D**,**E**), and 21 (**F**,**G**). Ventral section shows parts of median brain. High TH-lir in fibers in anterior and posterior medial protocerebrum neuropils and medial antennal neuropil (**A**). Section showing a decrease in TH-lir in fibers in posterior medial protocerebrum neuropil and medial antennal neuropil (**B**). Dorsal section showing a decrease in TH in AMPN, PMPN, CB, MAN, and LAN fibers at 7 days (**C**). Ventral section through median brain showing a decrease in TH-lir in fibers of anterior and posterior medial protocerebrum neuropils and medial antenna l neuropil at 14 days (**D**). Dorsal section showing a decrease in TH in AMPN, PMPN, CB, MAN, and LAN fibers at 14 (**E**) and 21 days (**F**). Decrease in TH-lir in fibers of anterior and posterior medial protocerebrum neuropils and medial antenna l neuropil at 21 days (**G**). (**H**) Changes in signal intensity of TH-lir in median brain after chronic exposure to low doses of rotenone. The letter designations are as follows: AMPN, anterior medial protocerebral neuropil; PMPN, posterior medial protocerebral neuropil; ON, olfactory neuropil; CB, central body; LAN, lateral antenna I neuropils; MAN, medial antenna l neuropils; PT, protocerebral tract; 6 and 11, cell clusters. Magenta color indicates TH; blue, DAPI. Scale bars: 100 μm. Data analysis was performed in GraphPad Prism 7. Each value is mean ± SEM (*n* = 6). Data were analyzed by two-way ANOVA followed by Dunnett’s post hoc test. The same letter indicates no significant difference (*p* > 0.05); the capital letter indicates a highly significant difference (*p* < 0.001).

After 14 days of rotenone treatment, a reduction in TH-lir was observed in AMPN, CB, PMPN, MAN, and LAN fibers (Figure 4D,E). After 21 days, TH-lir decreased in all regions of the brain, remaining only in a few small TH-lir neurons of cell clusters 6 and 11 and also in small nerve fibers (Figure 4F,G).

A quantitative analysis of corrected total cell fluorescence (CTCF) showed a significant decrease in the fluorescence signal in the median brain at 7, 14, and 21 days of rotenone treatment (to 1170.88 ± 79.97, 960.085 ± 74.75, and 824.565 ± 72.25, respectively), compared to the respective control values (2174.11 ± 130.62) (Figure 4H).

In the VNC of the control *H. sanguineus*, TH-lir was detected in all parts of the subesophageal (SEG), thoracic (TG), and abdominal ganglia (AG) (Figure 5A–E). In the SEG, TH-lir neurons (ranging from 10 to 30 μm in diameter) were present predominantly in dorsolateral clusters (DLC) (Figure 5A,B). The TG contained single TH-lir neurons in all cell clusters 22–27 and TH-positive fibers in the thoracic neuropils (Figure 5C). In addition, TH-lir was found along the midline and lateral fibers of the SEG and TG, and also around the large thoracic artery (Figure 5A,D). The abdominal ganglion (AG) contained numerous TH-lir fibers and few TH-lir neurons of 70–75 and 20–25 μm (Figure 5E,E1,E2).

After 7 days of rotenone treatment, the TH immunostaining revealed that the TH-lir loss affected all of the clusters indiscriminately and fibers in the VNC compared to the control (Figure 5F), except for several neurons in the AG (Figure 5F1). After 14 days, the VNC showed disappearance of TH-lir fibers in SEG, TG, and AG (Figure 5G). By day 21, the TH-lir loss affected all neurons and fibers of the VNC, with weak immunoreactivity detected only in cells (20–25 μm) of the AG (Figure 5H).

**Figure 5 ijms-25-07159-f005:**
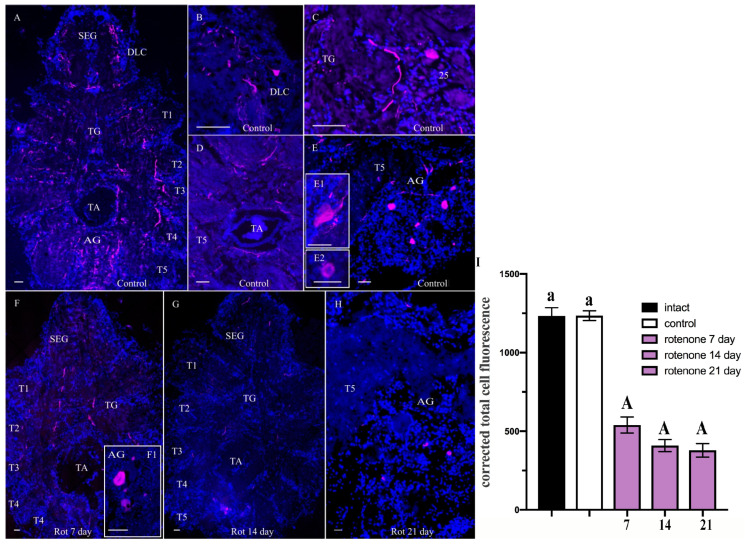
Changes in tyrosine hydroxylase (TH)-lir in ventral nerve cord (VNC) of control crabs *Hemigrapsus sanguineus* (**A**–**E**) and rotenone-treated animals at 7 (**F**), 14 (**G**), and 21 (**H**) days. Horizontal sections of VNC showing TH-lir neurons and nerve fibers in subesophageal ganglion (SEG), thoracic ganglia (T1–T5), and abdominal ganglion (AG) (**A**). A part of SEG showing immunoreactivity to TH-lir in dorsolateral clusters (DLC) (**B**). TH-lir in neurons and neuropil of TG (**C**). TH-lir in nerve fibers surrounding thoracic artery (TA) in TG (**D**). TH-lir neurons in AG (**E**), large TH-lir neuron (**E1**), small TH-lir neuron (**E2**). Dorsal section showing decrease in TH-lir in SEG, thoracic ganglia (T1–T5), and AG at 7 (**F**,**F1**), 14 (**G**), and 21 days (**H**). Changes in signal intensity of TH immunoreactivity in median brain after chronic exposure to low doses of rotenone (**I**). The letter designations are as follows: SEG, subesophageal ganglion; TG, thoracic ganglion; TA, thoracic artery; T1–T5, neuropils of TG; DLC, dorsolateral cluster; AG, abdominal ganglion; 25, cell cluster. Magenta color indicates TH; blue color, DAPI. Scale bars: 100 μm. Data analysis was performed in GraphPad Prism 7. Each value is mean ± SEM (*n* = 6). The qualitative immunostaining data were confirmed by calculations of CTCF. In the VNC, CTCF decreased compared to that of control (1235.12 ± 31.11) after 7 (539.367 ± 50.9), 14 (408.76 ± 38.77), and 21 days (378.525 ± 42.95) of rotenone treatment (**I**). Thus, the decrease was by 56.4%, 66.9%, and 69.3%, respectively, compared to the respective control values. Data were analyzed by two-way ANOVA followed by Dunnett’s post hoc test. The same letter indicates no significant difference (*p* > 0.05); the capital letter indicates a highly significant difference (*p* < 0.001).

#### 2.2.2. ChAT-lir after Chronic Exposure to Low Doses of Rotenone

In the median brain of the control *H. sanguineus*, ChAT-lir neurons were present in cluster 6, in single neurons in cell cluster 8, and also a very weak ChAT-lir was detected in the anterior (AMPN) and posterior (PMPN) medial protocerebral neuropils and in the protocerebral bridge (PB) neuropil (Figure 6A,B).

**Figure 6 ijms-25-07159-f006:**
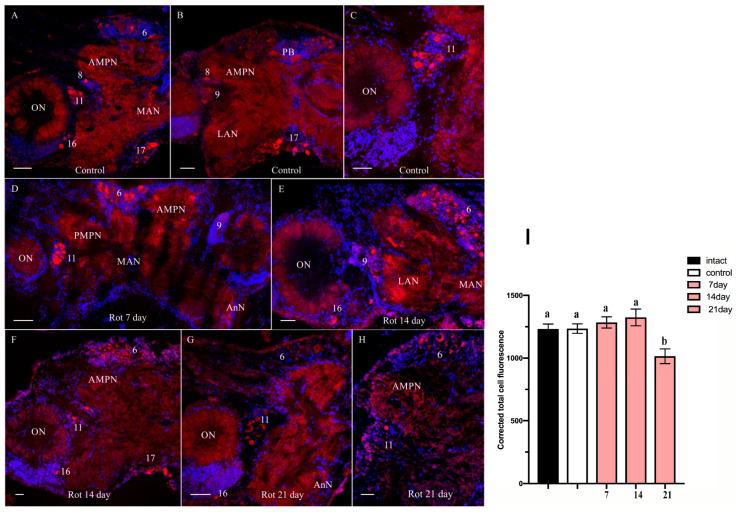
Changes in choline acetyltransferase (ChAT)-lir distribution in the median brain of control crabs *Hemigrapsus sanguineus* (**A**–**C**) and rotenone-treated animals at 7 (**D**), 14 (**E**,**F**), and 21 (**G**,**H**) days. The qualitative immunostaining data were confirmed by calculations of CTCF. All the treatment groups were compared to control using the one-way ANOVA test. As regards CTCF meaning, at 7 days of rotenone exposure, the results were as follows: F = 0.5205; *p* = ns; R^2^ = 0.06489. At 14 days: F = 1.109; *p* = ns; R^2^ = 0.1288. At 21 days: F = 7.501; *p* < 0.0055; R^2^ = 0.5. The data were analyzed using the *t*-test, which compared the experimental group at any given time with the control group. Data were analyzed by two-way ANOVA followed by Dunnett’s post hoc test. The same letter indicates no significant difference (*p* > 0.05); different letters indicate a significant difference (*p* < 0.05). Dorsal section showing parts of median brain. ChAT-lir in neuropils and neurons in cell clusters 6, 8, 11, 16, and 17 (**A**–**C**). Increase in ChAT-lir in cell cluster 6 (**E**,**F**). Decrease in ChAT-lir in cell clusters 6, 8, 11, 16, and 17 (**G**,**H**). Changes in signal intensity of ChAT-lir in median brain after chronic exposure to low doses of rotenone (**I**). The letter designations are as follows: AMPN, anterior medial protocerebral neuropil; PMPN, posterior medial protocerebral neuropil; ON, olfactory neuropil; PB, protocerebral bridge neuropil; LAN, lateral antenna I neuropils; MAN, medial antenna l neuropils; 6, 8, 9, 11, 16, and 17, cell clusters; ns, no significant difference. Red color indicates ChAT; blue color, DAPI. Scale bars: 100 μm. Data analysis was performed in GraphPad Prism 7. Each value is mean ± SEM (*n* = 6).

The size of labeled cells in cluster 6 varied within 12–42 μm; these were predominantly neurons 25–30 μm in diameter exhibiting different degrees of labeling. ChAT-positive neurons were identified also in the deutocerebrum clusters 9 and 11 (Figure 6A–C). Furthermore, ChAT-lir was detected also in the lateral antenna I neuropils (LAN), medial antenna I neuropil (MAN), and olfactory neuropils (ON), predominantly, on the periphery of the olfactory lobes. In the tritocerebrum, ChAT-lir was detected in medium-sized neurons (35–45 and 25–30 μm, respectively) of clusters 16 and 17 (Figure 6A,B).

From day 7 to 14 after the rotenone treatment, ChAT-lir insignificantly increased only in cluster 6 compared to those of the control crabs. In this group, small immunoreactive cells were labeled that were absent in the control (Figure 6D–F). A post hoc analysis showed that in the rotenone-treated crabs, ChAT-lir had a tendency to increase in the median brain compared to that in the control (Figure 6I). However, at 21 days after the experiment’s start, there was a decrease in ChAT-lir in all the regions of the median brain compared to the control (Figure 6G,H). In the rotenone-treated crabs, the CTCF values in the median brain decreased (1014.77 ± 58.66) compared to those in the control animals (1235.27 ± 37.86) (Figure 6I).

In the VNC of the control *H. sanguineus*, ChAT-lir was detected in the SG in neurons (with sizes varying within 15–48 μm) of dorsolateral clusters (Figure 7A). In the TG, ChAT-lir cells were present in all cell clusters 22–27 (Figure 7A,A1,B). Populations of large and medium-sized cells of this type were found near the thoracic artery, which runs through the middle of the ventral nerve cord (Figure 7B). In the AG, cell cluster 28 contained predominantly medium-sized (21–40 μm) ChAT-lir cells (Figure 7A,B). In all neuropils of the VNC, weak staining for ChAT was detected (Figure 7A,B).

**Figure 7 ijms-25-07159-f007:**
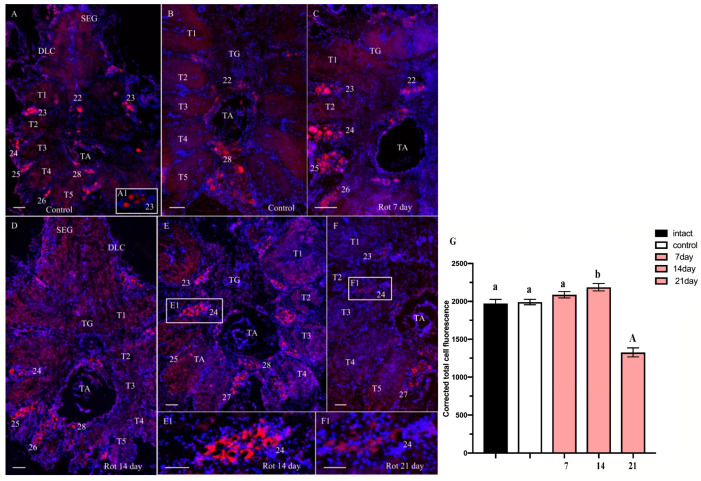
Changes in choline acetyltransferase (ChAT)-lir in ventral nerve cord (VNC) of control crabs *Hemigrapsus sanguineus* (**A**,**A1**,**B**) and rotenone-treated animals at 7 (**C**), 14 (**D**,**E**), and 21 (**F**) days. The qualitative immunostaining data were confirmed by calculations of CTCF. Data were analyzed by two-way ANOVA followed by Dunnett’s post hoc test. The same letter indicates no significant difference (*p* > 0.05); different letters indicate a significant difference (*p* < 0.05); the capital letter indicates a highly significant difference (*p* < 0.001). Dorsal section of VNC showing ChAT-lir neurons in subesophageal ganglion (SEG), thoracic ganglia (**A**,**A1**), and abdominal ganglion (AG) (**B**). Increase in ChAT-lir in TG cells clusters (**C**–**E**,**E1**). Decrease in ChAT-lir in thoracic ganglia (**F**,**F1**). Changes in signal intensity of ChAT immunoreactivity in VNC after chronic exposure to low doses of rotenone (**G**). The letter designations are as follows: SEG, subesophageal ganglion; TG, thoracic ganglion; TA, thoracic artery; T1–T5, neuropils of TG; DLC, dorsolateral cluster; AG, abdominal ganglion; 22–28, cell clusters. Scale bars: 100 μm. Red color indicates, ChAT; blue color, DAPI. Scale bars: 100 μm. Each value is mean ± SEM (*n* = 6).

From day 7 to 14 of rotenone exposure, a slight increase in ChAT-lir was observed in cell clusters 23–27 in the ventral regions of thoracic ganglia compared to those in the control (Figure 7C–E). However, by day 21, these cells clusters showed a decrease in ChAT-lir compared to the control (Figure 7F,F1). Furthermore, within the cells clusters of the VNC in the rotenone-treated crabs, there were signs of damage to neurons (Figure 7E1,F1).

A quantitative analysis of CTCF showed that the treatment with a low dose of rotenone significantly decreased the ChAT fluorescence intensity in the VNC (1326.34 ± 59.74) compared to that in the control VNC (1990.6 ± 36.08) (Figure 7G) at 21 days from the experiment’s start.

## 3. Discussion

In this study, we attempted to analyze the neurochemical and behavioral alterations caused by rotenone exposure in shore crabs. Chronic subcutaneous administration of rotenone at a low dose, similarly to Parkinson’s disease, causes TH to decrease in a number of CNS regions.

In the median brain of *H. sanguineus*, a progressive decrease in TH-lir was observed in neurons and fibers of the AMPN, MAN, and LAN, and then in the PMPN, the central body (CB), and the MAN to almost complete loss of TH-lir by day 21 after the rotenone treatment. In crustaceans, these anatomical regions have stereotyped projections to other brain regions (optic ganglia, lateral protocerebrum) and the VNC [38,39,40], but their role has not been fully elucidated. It is assumed that these brain structures, like in other arthropods, are implicated in sensory integration, motor coordination, and spatial orientation [39]. A decrease in TH-lir in these regions may cause impairment of movement control.

Also, our immunohistochemical data have shown the loss of TH enzyme labeling of the VNC, another CNS region that also seems to be vulnerable to the toxic effects of rotenone. In the VNC ganglia, the TH-lir content was reduced more significantly than in the median brain. Excessive loss of dopaminergic neurons in the VNC might lead to changes in the crab’s locomotor behavior. DA is a ubiquitous neuromodulator in the animal kingdom and is well known as an important signaling molecule for locomotor behavior [41,42,43]. In crustaceans, it provides dopaminergic modulation of the central pattern generator network of locomotion [44].

The effect of rotenone on the TH expression in decapod crustaceans has never been studied. However, there are reports that rotenone causes a decrease in dopaminergic neurotransmission in other groups of invertebrates [20]. Lack of TH immunolabeling is suggested to be a morphological indicator of dopaminergic cell loss during the induction of Parkinson’s symptoms in both vertebrates [45,46] and invertebrates such as *Drosophila* flies [3], the pond snail *Lymnaea stagnalis* [15], and the worm *Lumbricus terrestris* [5]. As behavioral experiments have shown, TH deficiency results in a reduction in locomotor activity and interference with behavior [3,15]. These effects progress quickly with acute exposure and more slowly during chronic exposure to lower, sublethal doses of rotenone [15].

In the present study, we found that after 7 days of chronic exposure to low doses of rotenone, animals still could move despite the low TH levels, but their locomotor activity reduced. Previously, the impaired locomotor behavior was recorded from other species of invertebrates and mammals in experiments where Parkinsonian symptoms were induced by rotenone and the animals showed hypokinetic behavioral signs in activity [3,20,47].

Besides the significantly decreased TH-lir in the rotenone-treated crabs compared to the control, we observed a variation in ChAT expression, which indicated that the toxicity of rotenone was not confined only to the dopaminergic system but it also exerted an effect on the cholinergic system. In *H. sanguineus*, the most pronounced variations in ChAT-lir were observed in VNC neurons. Cholinergic neurons *release acetylcholine* (*ACh*), an excitatory neurotransmitter important for motor control, and in the autonomic, enteric, and central nervous systems [25,48]. In both vertebrates and many invertebrates, ACh predominantly regulates the neuromuscular transmission [25,49]. In such crustaceans as crayfish, ACh application in the VNC can induce rhythmic activity in the walking leg motor neuron pools, and focal application even elicits differential changes in membrane potential in those motor neurons [50]. Cholinergic agonists both activate and modulate the fictive swimmeret motor pattern [51,52,53,54]. Moreover, ACh is a neurotransmitter of the coordinating neurons in the VNC [55].

A slight increase in ChAT in some of the brain regions and the VNC is a well-known phenomenon. There was an increase in ChAT activity in the mammal brain when a compensatory capability of the cholinergic system and adaptation to new circumstances were demonstrated [56].

With long-term exposure of crabs to low doses of rotenone, a significant decrease in ChAT-lir was documented. Several earlier studies showed that the activity of ChAT-positive neurons may decrease in PD [14]. As the authors noted, the decrease in the number of cells with immunohistochemically detected ChAT expression was preceded by lower expression of TH in all regions. In mammals, the striatal cholinergic interneuron activity and ACh release were also reported to be modulated by DA neurons [13], and the ACh neurotransmission system is suppressed when the DA system becomes dysfunctional [10].

At 21 days after the experiment’s start, rotenone-treated crabs showed signs of damage to ChAT-lir neurons in the VNC. The mortality of rotenone-treated crabs at 21 days allows an assumption that the ChAT decrease may be involved in the decreased locomotor activity of the animals and their subsequent death. These findings are consistent with the previously reported consideration of rotenone as one of stimuli which may play a role in neurodegeneration of cholinergic neurons in the mammalian brain [57,58]. Furthermore, there is evidence that rotenone induces cell death of cholinergic and dopaminergic neurons in an organotypic cell culture model [58].

The mechanism that leads to cell death after rotenone treatment is not fully understood. Rotenone is a highly potent reference complex I inhibitor [57,59,60] that can trigger oxidative stress damage in the CNS. Complex I inhibitors affect NADH-ubiquinone reductase, which is the first and the largest enzyme of the five complexes of the mitochondrial respiratory electron transport chain and also a major source of reactive oxygen species (free radicals) during cellular respiration [2,61]. When the complex I functions are impaired, the oxidizing effect of free radicals (generally regarded as oxidative stress) triggers a cascade of degenerative intracellular processes, potentially leading to cell death [4,62,63].

The present study has demonstrated that chronic exposure to low concentrations of rotenone can lead to the neurotoxic effect on dopaminergic and cholinergic neurons in the brain and VNC, which interferes with the normal status of the CNS. Dopamine and acetylcholine are important neurotransmitters involved in motor regulations; variations in the levels of these neurotransmitters result in motor deficits and changes in locomotor behavior. Furthermore, long-term exposure to low concentrations of rotenone may cause cell death in animals. In our study, the chronic rotenone treatment decreased the DA and ACh levels in the brain and VNC. It also led to the progressive and irreversible reduction in locomotion, life span of the treated crabs, and changes in their behavior.

Thus, according to the results of our study as regards the effects of rotenone on *H. sanguineus*, rotenone-treated shore crabs share many common features with widely used models of Parkinson’s disease [both vertebrate and invertebrate] [64]. The rotenone-induced neurotransmitter modifications and associated influence on behavior can be an instrumental approach to unveiling the mechanism by which rotenone causes neurotoxicity and to understanding the cellular mechanisms involved in neurodegenerative diseases influenced by this environmental pollutant.

## 4. Materials and Methods

### 4.1. Experimental Animals

Adult males of the shore crab *Hemigrapsus sanguineus* (de Haan, 1835) (Varunidae, Decapoda) with carapace widths of 40–45 mm were caught in Peter the Great Bay, Sea of Japan. The animals were kept in 140 L glass tanks (pH 7.0–7.6, at room temperature of 18 °C) with constantly aerated and filtered natural seawater (salinity 32‰) under a 12 h/12 h light/dark cycle. The animals were fed once a day with frozen squid purchased at a market, but were deprived of food on the experiment day. All the animals were acclimated to the laboratory conditions for a week before the experiments.

After the acclimation, the animals were divided into three groups: intact animals, experimental group, and control group. The animals of the experimental group were injected with a rotenone solution (ROT) at a concentration of 1.25 µg/L into the left cheliped. The control animals were injected with the same volume of a 0.9% NaCl solution. The intact group animals did not receive any treatment. The injections were made once every two days for two weeks. The stock solution of ROT (Sigma-Aldrich, St. Louis, MO, USA) was prepared at a concentration of 5 mg/mL in dimethyl sulfoxide. The ROT concentration was selected based on our explorative experiment.

Behavioral assessment of the experimental and control animals was performed at 7, 14, 21, and 28 days after the experiment’s start. In parallel, we collected tissue samples for immunohistochemistry. The duration of the experiment was 28 days. Nervous ganglia from the intact, control, and rotenone-treated groups were sampled and dissected at the same time.

All applicable international, national, and/or institutional guidelines for the care and use of animals were carefully adhered to. All possible effort was made to minimize the number of animals used.

### 4.2. Behavioral Assessment

#### Open Field Test

Locomotor activity was measured in an open-field test. Glass tanks (0.85 × 0.56 × 0.47 m) were used as the open field. For all trials, each crab was placed at the center of the open field. The crabs could move freely, with their movements recorded during 5 min. Each trial was recorded with a web camera positioned directly above the open field. The behavioral test was performed at 7, 14, 21, and 28 days for each group. Locomotor behavior was assessed on the basis of the total distance moved (cm).

### 4.3. Antibodies

Rabbit anti-TH antibody (EDM Millipore, Cat. No. AB152) targets TH as a key enzyme involved in tyrosine biosynthesis. Tyrosine hydroxylase (TH) was previously identified in ganglia of arthropods, including crustaceans [65,66].

Previous immunohistological studies on related crustaceans demonstrated that antibodies against DA and TH yield highly consistent staining patterns [31,67], thereby validating the use of TH antibody as a reliable marker of dopaminergic neurons in crustaceans.

Cytoplasmic choline acetyltransferase (ChAT), which synthesizes acetylcholinesterase (AChE), is a much more specific marker of cholinergic neurons than AChE. In the present study, we used the concentration of antibody to ChAT recommended by the manufacturer, along with a concentrated blocking buffer to eliminate non-specific binding. Antibodies against these proteins are used as phenotypic markers for cholinergic neurons [68].

To visualize neuropils, some of the sections were incubated with mouse monoclonal anti-synapsin antibody, which targets a presynaptic marker (SYNORF1 or antibody 3C11). As a previous study showed, this antibody detects an epitope widely conserved in the nervous systems of arthropods, including crustaceans [69,70].

### 4.4. Immunocytochemistry

The brain and VNC were rapidly dissected on ice and fixed with a 4% paraformaldehyde (PFA) solution in phosphate-buffered saline (PBS) (pH 7.4) at 4 °C for 2 h. The fixed samples were washed in PBS five times at 4 °C for 2 h, cryoprotected by incubation at 4 °C overnight in PBS containing 15% sucrose, then embedded in medium with optimal cutting temperature, and frozen at −20 °C. Sections 25–30 μm were mounted on slides, coated with poly-L-lysine (Sigma, USA), air-dried, and stored at −20 °C for subsequent staining. For immunohistochemical staining, frozen sections were processed as described previously [71].

To eliminate non-specific binding, the samples were incubated in the blocking buffer (5–10% normal donkey serum (Jackson ImmunoResearch, Baltimore Pike, PA, USA), 1–2% Triton-X 100 (Sigma), and 1% bovine serum albumin (Sigma) in 1 × PBS) overnight at 4 °C, with the primary antibody diluted in this blocking buffer. The samples were then incubated overnight at 4 °C with a primary rabbit anti-TH antibody (1:2000 (EMD Millipore, Billerica, MA, USA) or goat anti-ChAT polyclonal antibody (1:500) (Millipore, Burlington, MA, USA).

In some of the experiments, primary antibody against synapsin (1:500, clone 3C11, DSHB) was also used to stain neuropils as described earlier [69,70]. After being washed in 0.01 M PBS (pH 7.4) containing 0.5% Triton X-100 (PBST, pH 7.4), the sections were incubated with donkey secondary antibody conjugated with 488-, 555-, or 647-Alexa Fluor (1:1000; Invitrogen, Thermo Fisher Scientific, Waltham, MA, USA) in combination with a nuclear marker, DAPI (4′,6-diamidino-2-phenylindole) (Sigma-Aldrich; Millipore, Burlington, MA, USA), diluted to 10 mg/mL for 2 h at 22 °C. Then, the sections were washed with PBS and embedded in glycerol (Merck, Kenilworth, NJ, USA).

### 4.5. Microscopy and Imaging

The immunostained sections were examined and images for analysis were taken using a confocal microscope Zeiss LSM 700 at the Far Eastern Center of Electron Microscopy, A.V. Zhirmunsky National Scientific Center of Marine Biology, Far Eastern Branch, Russian Academy of Sciences, Vladivostok, Russia. All images were processed and analyzed using the Imaris (Bitplane, Zurich, Switzerland) and ImageJ 1.53 (National Institutes of Health, Bethesda, MD, USA) software. The presented figures show the projections of maximum immunoreactivity.

### 4.6. Quantification and Statistical Analysis

All values are represented as mean ± SEM. The “N” value refers to the number of animals used for each experiment (the biological N). When only two groups were compared, Student’s *t*-test was used. If more than two groups were compared, *p* values were estimated by the one-way analysis of variance (ANOVA) followed by Dunnett’s multiple-comparisons test. The actual *p* values and the number of samples are provided in the text. Samples were analyzed blindly to experimental conditions. All samples were included, with no exclusions. The data were analyzed with one- or two-way ANOVA. We followed the methodical and statistical analysis patterns of the following articles [72,73].

The survival rates for the control and intact groups were 100% during a 4-week experimental period. The effectiveness of the different procedures within a treatment was compared with the help of survival curves (Figure 2). The ‘survdiff’ function (‘survival’ package) from R version 3.6.1. [74] was used to calculate chi-square distance and the corresponding *p*-value [75,76]. We compared each curve within a treatment type following pairwise multiple comparisons with Bonferroni correction in the case of significant difference among the curves (Table 1). The overall chi-square statistic is 15 with 2 degrees of freedom and a *p*-value of 0.0005, indicating significant differences in survival between the groups.

The series of sections from the brain and VNC were examined. To quantify fluorescence, images were taken for control, intact, and rotenone-treated animals (7, 14, 21, and 28 days) with the same scan settings. We here use the terms “staining intensity” and “level” to describe the relative amounts of immunoreactive staining in positive cells or fibers when visually compared to the control. When these terms are used to describe immunoreactive staining of different antigens, they refer to the fluorescence intensity resulting from the immunochemical procedures. The brain and VNC sections used for immunohistochemistry and quantitative estimation were cut strictly from the same regions. This allowed us to accurately compare the distributions of immunoreactive neuromolecules in tissues. For fluorescence intensity quantification, the images of the immunostained neuronal tissues were processed using the ImageJ 1.51w image processing software [77,78]. Each photograph was imported into ImageJ, and the images were adjusted to the threshold using the method of Otsu (1979) [79]; the area and mean fluorescence of the foreground (signal of Alexa 488) and background were measured across the image. The corrected fluorescence was calculated using the Hammond’s formula (QBI, The University of Queensland, Australia, https://theolb.readthedocs.io/en/latest/imaging/measuringcellfluorescenceusingimagej.html) from 10 August 2021.

The corrected total cell fluorescence (CTCF) signal = mean fluorescence of foreground − (foreground area × mean fluorescence of background).

The areas expressing specific molecular markers were measured in representative sections from both the brain and the VNC of at least six crabs. All quantifications were performed using ImageJ, and the values were processed using the Prism 7 software (GraphPad, San Diego, CA, USA).

### 4.7. Neuroanatomical Nomenclature

The neuroanatomical nomenclature used in this study is based on that proposed by Sandeman and co-authors [37] with some modifications adopted from the studies of Harzsch and Hansson [69] and Richter with co-authors [80]. Also, we used the classification of cell clusters (numbered 1–17 from anterior to posterior) in the shore crab brain and the standard nomenclature developed in previous studies for these cell clusters, as well as for neuropils and regions of the ventral nerve cord (VNC) described from several decapod species [81,82,83].

## Figures and Tables

**Figure 1 ijms-25-07159-f001:**
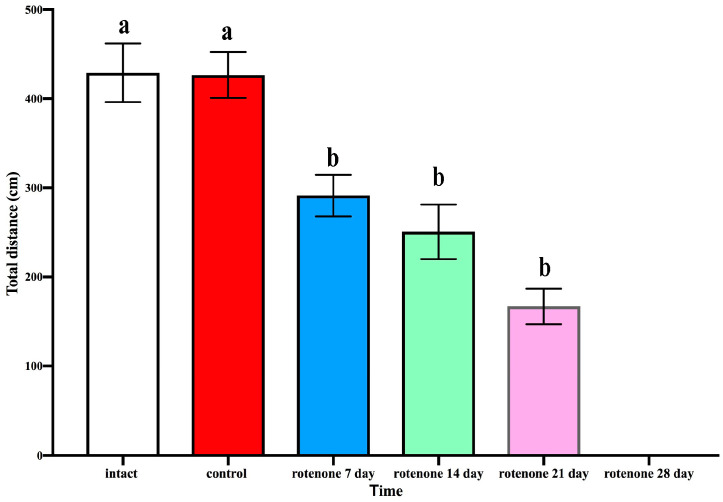
Behavioral open field test. Distance that crabs *H. sanguineus* moved during 5 min in control and after rotenone treatment. Each value is mean ± SEM (*n* = 6). The distance covered decreased significantly already from day 7 after the beginning of rotenone treatment. Significant decreases in locomotor activity also became evident on day 7 (*p* < 0.001). The significant reduction continued throughout the treatment period. On day 21 after the experiment’s start, the crabs in the rotenone group covered an average distance of 166.917 ± 19.90 cm for 5 min, which corresponded to a decrease in the locomotor activity by 39.1% (*p* < 0.001) compared to that in control animals between days 22 and 26, the rotenone-treated crabs died. The data were analyzed using the *t*-test, which compared the experimental group at any given time with the control group. The same letter indicates no significant difference (*p* > 0.05); different letters indicate a significant difference (*p* < 0.05).

**Figure 2 ijms-25-07159-f002:**
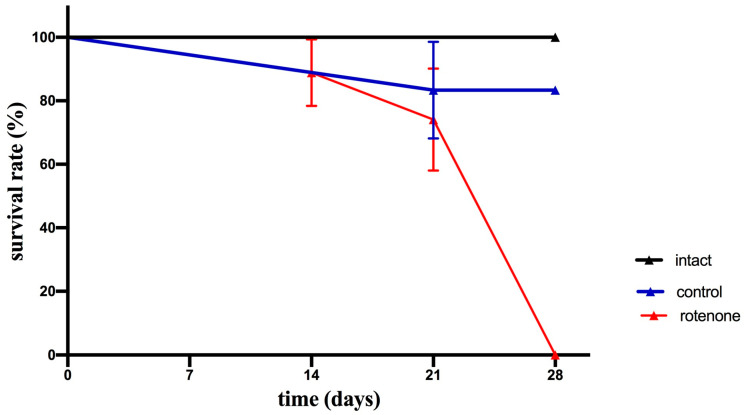
Survival rates of intact crabs *H. sanguineus*, in control, and after treatment with rotenone for 28 days. Each value is mean ± SEM (*n* = 6). After 28 days, the rotenone-treated crabs died.

**Table 1 ijms-25-07159-t001:** Pairwise comparison of different treatments: χ^2^ and *p*-values (degrees of freedom = 2).

Group	N	Observed	Expected	(0 − E)^2/E^	(0 − E)^2/V^
Control	23	1	1.950	0.463	0.798
Intact	24	0	2.199	2.199	4.130
Rotenone	11	4	0.851	11.655	14.744

Chi-square statistic: 15 on 2 degrees of freedom, *p*-value: 0.005.

## Data Availability

The datasets used and analyzed during the current study are available from the corresponding author on reasonable request.

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
