# Peer review of "Effects of Chronic Exposure to Low Doses of Rotenone on Dopaminergic and Cholinergic Neurons in the CNS of *Hemigrapsus sanguineus"

_ijms, 2024, doi:10.3390/ijms25137159_

Round 1

Reviewer 1 Report

Comments and Suggestions for Authors

Review of “Effects of Chronic Exposure to LOW Doses of Rotenone on Dopaminergic and Cholinergic Neurons in the CNS of Hemigrapsus sanguineus”

1.      The open filed test for crabs is interesting. It would be interesting to know more how it was conducted with lighting and housing facilities. Was natural sea water used  or made from sea salts? Temp ?

Do females and males move as same rate ? Is one sex larger than the other ? I see that only males are used in this study . Is that due to size differences in the males and females ?

 2.      Figure 1 is a little confusing. Should there not be a control line the same amount of days as the experimental? Maybe there is fatigue or familiarity with the environment and not wanting to keep exploring after 21 days ? So, running a control along with the experimental for the same number of days could also show a decrease in movement. A control group is used in Figure 2 for survival.

 3.      The  immunoreactivity staining in all the graphs looks fine. Nice figures. The study is explained well and the methods detailed enough for one to be able repeat  if needed or to build on for future studies.

 4.      One point the authors could make is that they injected  rotenone and the animals were not subject water containing rotenone as would be expected in a natural setting. So, would rotenone naturally be taken up by the gills or digestion from eating compounds containing rotenone? Also, would rotenone cross the epithelia lining in the crab  GI track ?

 5.      Are Hemigraphsus sanguineus models comparable to human physiology in some way? Would like to see some literature included in introduction on why this specific species was chosen for study.

 6.      There are some formatting issues with the font style and size changing every once in a while in the middle of sentences...

 7.      Figure 1 should have three separate lines: one for control, one for rotenone exposure, and one for intact group. Furthermore, what does the author mean by "abnormalities of orientation" in the paragraph immediately following figure 1?

 8.      Sentence seems prematurely ended in first paragraph following figure 5, "Horizontal sections of VNC."

 9.      Is p<0.1 significant enough to note? See figure 6.

Author Response

Comments and Suggestions for Authors

Review of “Effects of Chronic Exposure to LOW Doses of Rotenone on Dopaminergic and Cholinergic Neurons in the CNS of Hemigrapsus sanguineus”

  1. The open filed test for crabs is interesting. It would be interesting to know more how it was conducted with lighting and housing facilities. Was natural sea water used or made from sea salts? Temp ?

Reply

During the adaptation period and throughout the experiment, the room conditions were maintained, with a room temperature of 18°C and a 12/12 h light/dark cycle (light on 08:00–20:00 h). The animals were kept in 140 L tanks (pH 7.0-7.6) with constantly aerated and filtered natural sea water. The sea water was supplied from the bay. The open field test was conducted in a controlled environment with low noise and under fluorescent lamps positioned in such a way to avoid shadows.

Do females and males move as same rate ? Is one sex larger than the other ? I see that only males are used in this study . Is that due to size differences in the males and females ?

Reply

Shore crabs Hemigrapsus sanguineus have pronounced sexual dimorphism. Males generally have larger body and claw sizes, whereas females have larger abdomens. In this experiment, we did not study sex-related differences in sensitivity to rotenone. This remains to be clarified in the future, since the sensitivity of organism to the toxicant may depend, among other things, on its hormonal background (especially during the breeding season). The experiment was conducted in the spring and summer seasons, when female shore crabs contain eggs. Adhering to the principle of humanity, we did not take females with eggs for the experiments.

  1. Figure 1 is a little confusing. Should there not be a control line the same amount of days as the experimental? Maybe there is fatigue or familiarity with the environment and not wanting to keep exploring after 21 days ? So, running a control along with the experimental for the same number of days could also show a decrease in movement. A control group is used in Figure 2 for survival.

Reply

Figure 1 has been changed according to the comment (the control line shows the same number of days as the experimental line). On day 28 after the experiment’s start, the distance moved is zero, because the rotenone-treated crabs died between days 22 and 26.

3.The immunoreactivity staining in all the graphs looks fine. Nice figures. The study is explained well and the methods detailed enough for one to be able repeat if needed or to build on for future studies.

  1. One point the authors could make is that they injected rotenone and the animals were not subject water containing rotenone as would be expected in a natural setting. So, would rotenone naturally be taken up by the gills or digestion from eating compounds containing rotenone? Also, would rotenone cross the epithelia lining in the crab GI track?

Reply

It has been repeatedly shown that in natural conditions rotenone, after entering an aquatic environment, is absorbed through the gills, the skin, and the digestive tract in several aquatic invertebrates and fishes. Its toxic effect depends on the dose, time of exposure, ambient temperature, illumination, etc. In natural conditions, rotenone can enter the crab’s body through the gills and the digestive tract. Therefore, in experimental studies of aquatic animals, rotenone treatment can be provided by dissolution of the compound in the water (containing < 0.01% DMSO as a solvent) where they live. However, in shore crabs, this model poses a challenge in estimating the proper dose and time of rotenone exposure. In chronic experiments, lethality was avoided by using rotenone injections in a small single dose. The use of injections allows accurate estimation of the dose of the drug to be injected into the animal's body per gram of crab weight and is widely used to study effects of hormones, neurotoxins, etc. in crabs (Kornthong et al., 2013; 2014). Systemic injections of rotenone, which has high lipophilicity and, therefore, penetrates all cell membranes and enters into all organs, including the central nervous system, make it possible to assess its effect on the activity of neurotransmitters.

  1. Are Hemigraphsus sanguineus models comparable to human physiology in some way? Would like to see some literature included in introduction on why this specific species was chosen for study.

Reply

The Asian shore crab, Hemigrapsus sanguineus (de Haan, 1835), being an abundant and widespread inhabitant of coastal and estuarine waters, is highly tolerant to a wide range of temperatures and salinities, which characterizes this species as highly adapted to changing environmental conditions. Although crabs are evolutionarily distant from humans, fundamental cellular processes, as well as some regulatory functions of neurotransmitters and signaling pathways, are conserved between both organisms. Furthermore, the fundamental aspects of locomotor behavior regulation in decapod crustaceans, as in other arthropods, are quite similar to those in vertebrates, in particular as regards the structural and functional organization of the ventral nerve cord (VNC) that controls most of automated, rhythmically occurring processes. Prolonged systemic administration of rotenone makes it possible to bring the model closer to the formation of a chronic neurodegenerative process, which is consistent with the classic course of Parkinson's disease (PD). This crab may be a suitable model for the study of the neural organization of motor control and the cellular and molecular mechanisms of development of pathological processes in nervous tissues, including those in PD.

  1. There are some formatting issues with the font style and size changing every once in a while in the middle of sentences...

Reply

The font style and size have been corrected

  1. Figure 1 should have three separate lines: one for control, one for rotenone exposure, and one for intact group. Furthermore, what does the author mean by "abnormalities of orientation" in the paragraph immediately following figure 1?

Reply

Figure 1 has been redesigned. The sentence with …." abnormalities of orientation"…has been deleted.

  1. Sentence seems prematurely ended in first paragraph following figure 5, "Horizontal sections of VNC."

Reply

The sentence «…Horizontal sections of VNC. Dorsal section of VNC showing TH-lir neurons» has been changed for «     Horizontal sections of VNC showing TH-lir neurons»

  1. Is p<0.1 significant enough to note? See figure 6.

Reply

No, it is not, you are right. We have deleted it.

Thank you so much

Elena Kotsyuba

Vyacheslav Dyachuk

Reviewer 2 Report

Comments and Suggestions for Authors

Introduction:

- The sentence on lines 53-54 should not be formatted in bold.

- At the first mention of a species, the authors should include both the author and the date of its description.

Material & methods:

The statistical analysis section is currently poorly designed and described. Repeatability is a fundamental aspect of scientific manuscripts. To address this, please describe and perform tests for normal distribution (e.g., Shapiro-Wilk test) and test for homogeneity of variance (e.g., Levene's test). If your data meet the assumptions of these tests, you can proceed with parametric tests. If not, you will need to perform non-parametric tests. Ensure that you describe these tests thoroughly.

I recommend checking and citing the following research as examples for how to perform and describe the above-mentioned statistical tests in a similar subject article:
https://www.sciencedirect.com/science/article/pii/S1532045622001818?via%3Dihub
https://www.mdpi.com/2305-6304/9/6/125

Results: 

Figure 1 requires a complete revision. The independent measured data should not be connected with lines, as this implies a linkage that does not exist. Instead, I recommend using bar diagrams to represent the data. Additionally, perform statistical tests to compare the results, and indicate the differences using letters, as demonstrated in the previously mentioned articles.

Figure 2 should be revised to start the line from day 0. Additionally, perform a survival analysis (e.g., using the survdiff function in R) to check for significant differences between survival lines.

For guidance on performing and illustrating survival analysis, I recommend reviewing the following articles:

https://www.cambridge.org/core/journals/journal-of-helminthology/article/abs/digenean-holostephanus-trematoda-digenea-cyathocotylidae-metacercariae-in-common-carp-cyprinus-carpio-linnaeus-1758-muscle-zoonotic-potential-and-sensitivity-to-physicochemical-treatments/6514457492F25713B1AAC7B2D07BE468?utm_campaign=shareaholic&utm_medium=copy_link&utm_source=bookmark

http://www.zoonotes.bio.uni-plovdiv.bg/ZooNotes_2024/ZooNotes_238_2024_Yancheva%20et%20al..pdf

For the remaining bar diagrams (Fig 4-7), I recommend highlighting the significant differences with small letters, as demonstrated in the articles mentioned.

I also recommend highlighting the significant differences with small letters for the remaining bar diagrams, as demonstrated in the previously mentioned articles. Additionally, please add the values of the statistical tests in the results section. Refer to the previously linked articles for guidance.

Author Response

Comments and Suggestions for Authors

Introduction:

- The sentence on lines 53-54 should not be formatted in bold.

Reply

The format of the font in the sentence on lines 53-54 has been corrected.

- At the first mention of a species, the authors should include both the author and the date of its description.

Reply

The proposed change has been made to the text: Asian shore crab Hemigrapsus sanguineus «(de Haan, 1835)»

Material & methods:

The statistical analysis section is currently poorly designed and described. Repeatability is a fundamental aspect of scientific manuscripts. To address this, please describe and perform tests for normal distribution (e.g., Shapiro-Wilk test) and test for homogeneity of variance (e.g., Levene's test). If your data meet the assumptions of these tests, you can proceed with parametric tests. If not, you will need to perform non-parametric tests. Ensure that you describe these tests thoroughly.

I recommend checking and citing the following research as examples for how to perform and describe the above-mentioned statistical tests in a similar subject article: https://www.sciencedirect.com/science/article/pii/S1532045622001818?via%3Dihub https://www.mdpi.com/2305-6304/9/6/125

Reply

We have added information about the statistical analysis to Materials and Methods and the values in the captions to the figures accordingly.

Results:

Figure 1 requires a complete revision. The independent measured data should not be connected with lines, as this implies a linkage that does not exist. Instead, I recommend using bar diagrams to represent the data. Additionally, perform statistical tests to compare the results, and indicate the differences using letters, as demonstrated in the previously mentioned articles.

Reply

According to the recommendation on Figure 1, we used bar diagrams to represent the data.

Figure 2 should be revised to start the line from day 0. Additionally, perform a survival analysis (e.g., using the survdiff function in R) to check for significant differences between survival lines.

Reply

Figure 2 has been revised to start the line from day 0.

For guidance on performing and illustrating survival analysis, I recommend reviewing the following articles: https://www.cambridge.org/core/journals/journal-of-helminthology/article/abs/digenean-holostephanus-trematoda-digenea-cyathocotylidae-metacercariae-in-common-carp-cyprinus-carpio-linnaeus-1758-muscle-zoonotic-potential-and-sensitivity-to-physicochemical-treatments/6514457492F25713B1AAC7B2D07BE468?utm_campaign=shareaholic&utm_medium=copy_link&utm_source=bookmark

http://www.zoonotes.bio.uni-plovdiv.bg/ZooNotes_2024/ZooNotes_238_2024_Yancheva%20et%20al..pdf

For the remaining bar diagrams (Fig 4-7), I recommend highlighting the significant differences with small letters, as demonstrated in the articles mentioned. I also recommend highlighting the significant differences with small letters for the remaining bar diagrams, as demonstrated in the previously mentioned articles. Additionally, please add the values of the statistical tests in the results section. Refer to the previously linked articles for guidance.

Reply

Thank you so much for the recommendations on the design of statistics and links to articles that we have carefully studied. We tried to use letters for the statistical difference. Thanks to you, these are better represented in the new version of the manuscript

Thank you so much

Elena Kotsyuba

Vyacheslav Dyachuk

Round 2

Reviewer 2 Report

Comments and Suggestions for Authors

The authors performed the main part of the reviewers’ comments adequately. However, for the statistical tests, they followed the guidance and patterns of the recommended articles but did not cite them in the respective sentences, such as in the Materials and Methods section. Therefore, I recommend including a sentence like this:

"We followed the methodical and statistical analysis patterns of the following articles: 
https://www.sciencedirect.com/science/article/pii/S1532045622001818?via%3Dihub
https://www.mdpi.com/2305-6304/9/6/125"

Fig.2.:
Furthermore, while the authors stated that they tried to improve the survival analysis with a statistical method, they did not actually perform it. A statistical survival analysis is crucial to determine whether there are significant differences among the survival rates. As noted in the previous review:

I copy paste here the part of the previous review: 

"Additionally, perform a survival analysis (e.g., using the survdiff function in R) to check for significant differences between survival lines.

For guidance on performing and illustrating survival analysis, I recommend reviewing the following articles: https://www.cambridge.org/core/journals/journal-of-helminthology/article/abs/digenean-holostephanus-trematoda-digenea-cyathocotylidae-metacercariae-in-common-carp-cyprinus-carpio-linnaeus-1758-muscle-zoonotic-potential-and-sensitivity-to-physicochemical-treatments/6514457492F25713B1AAC7B2D07BE468?utm_campaign=shareaholic&utm_medium=copy_link&utm_source=bookmark

http://www.zoonotes.bio.uni-plovdiv.bg/ZooNotes_2024/ZooNotes_238_2024_Yancheva%20et%20al..pdf"

Therefore, it is strongly recommended to include a comprehensive statistical survival analysis to assess the real differences among the survival rates.

Author Response

Comments and Suggestions for Authors

The authors performed the main part of the reviewers’ comments adequately. However, for the statistical tests, they followed the guidance and patterns of the recommended articles but did not cite them in the respective sentences, such as in the Materials and Methods section. Therefore, I recommend including a sentence like this:

"We followed the methodical and statistical analysis patterns of the following articles:

https://www.sciencedirect.com/science/article/pii/S1532045622001818?via%3Dihub

https://www.mdpi.com/2305-6304/9/6/125"

Reply

Sorry, we added suggested phrase and these journal links.

Fig.2.:

Furthermore, while the authors stated that they tried to improve the survival analysis with a statistical method, they did not actually perform it. A statistical survival analysis is crucial to determine whether there are significant differences among the survival rates. As noted in the previous review:

I copy paste here the part of the previous review:

"Additionally, perform a survival analysis (e.g., using the survdiff function in R) to check for significant differences between survival lines.

For guidance on performing and illustrating survival analysis, I recommend reviewing the following articles: https://www.cambridge.org/core/journals/journal-of-helminthology/article/abs/digenean-holostephanus-trematoda-digenea-cyathocotylidae-metacercariae-in-common-carp-cyprinus-carpio-linnaeus-1758-muscle-zoonotic-potential-and-sensitivity-to-physicochemical-treatments/6514457492F25713B1AAC7B2D07BE468?utm_campaign=shareaholic&utm_medium=copy_link&utm_source=bookmark

http://www.zoonotes.bio.uni-plovdiv.bg/ZooNotes_2024/ZooNotes_238_2024_Yancheva%20et%20al..pdf"

Therefore, it is strongly recommended to include a comprehensive statistical survival analysis to assess the real differences among the survival rates.

 Reply

We have performed statistical survival analysis according recomendation and described Procedures used for the statistical analysis from suggested articles (Sándor et al., 2019). The ‘survdiff’ function (‘survival’ package) from R version 3.6.1. (R Core Team, 2015) was used to calculate chi-square distance and the corresponding P-value (Harrington & Fleming, 1982).
